# DATASET DISTILLERS ARE GOOD LABEL DENOISERS

## ABSTRACT

Dataset distillation aims to synthesize a small set of informative samples that preserve the generalization ability of large datasets. However, its behavior under noisy conditions remains underexplored. In this paper, we systematically study dataset distillation under three representative noise types: symmetric, asymmetric, and natural noise. We first discover that, when the noise ratio exceeds a critical threshold, mainstream distillation methods consistently outperform training on the full noisy dataset using significantly fewer distilled samples. In contrast, under asymmetric noise, the structured label corruption often entangles with semantic features, making it difficult for distilled samples to recover the clean data distribution. We further validate the effectiveness of dataset distillation on real-world noisy datasets, highlighting its robustness under high noise but degraded performance in low-noise settings due to over-compression. To provide theoretical insights, we derive upper and lower bounds on the required images per class (IPC) under each noise type, grounded in information theory and PAC-Bayes analysis. Our findings offer both empirical and theoretical guidelines for effective distillation in noisy learning scenarios.

## 1 INTRODUCTION

Dataset distillation seeks to synthesize a small set of informative samples—typically a few *images per class* (IPC)—that can train a model to match the performance of full-dataset training Wang et al. (2018a). This paradigm has shown promise in reducing storage, accelerating training, and enabling data sharing under privacy constraints. While prior studies focus primarily on distillation from clean datasets Zhao & Bilen (2021); Cazenavette et al. (2022), its behavior under label noise remains largely unexplored.

Label noise, however, is ubiquitous in real-world settings due to annotation errors, semantic ambiguity, or automatic labeling pipelines. Classical approaches to noisy-label learning often rely on estimating the noise distribution and applying filtering Han et al. (2018); Malach & Shalev-Shwartz (2018); Wang et al. (2018b), reweighting Shu et al. (2019); Ren et al. (2019); Zhou et al. (2024), or relabeling strategies Li et al. (2023); Liu et al. (2022). These methods typically rely on accurate estimates of sample-level noise confidence, which can be difficult to obtain in real-world or high-noise regimes. Moreover, as observed in Ciortan et al. (2021); Yao et al. (2021), such iterative pipelines risk entering a vicious feedback loop: poor initial noise estimation leads to misdirected correction, which in turn reinforces flawed assumptions.

Rather than attempting to identify or filter out noisy labels explicitly, a promising alternative is to distill a support set that best spans the manifold of clean semantic representations Wang et al. (2018a). This concept aligns with the core principle of dataset distillation: to synthesize a minimal set of synthetic samples that encode task-relevant inductive bias Zhao & Bilen (2021); Cazenavette et al. (2022). While standard deep networks are prone to memorizing both clean and noisy patterns Arpit et al. (2017); Han et al. (2018); Yu et al. (2019), dataset distillation—by necessity—prioritizes consensus patterns that generalize well. Thus, we hypothesize that distilled samples may naturally suppress outliers and label noise by virtue of compression, a property we aim to rigorously test and formalize.

This observation raises a key question: *Can dataset distillation inherently act as a denoising mechanism, by retaining consistent semantic structure while suppressing noisy signals?*

In this paper, we analyze the robustness of dataset distillation under three canonical noise regimes: (i) **Symmetric noise**, where labels are uniformly randomized; (ii) **Asymmetric noise**, where corruption occurs between semantically similar classes; (iii) **Natural noise**, derived from human annotations in datasets like CIFAR-10N/100N Wei et al. (2021).

We study three representative distillation methods—DATM Guo et al. (2023) (parameter matching), DANCE Zhang et al. (2024) (distribution matching), and RCIG Loo et al. (2023) (meta-learning)—and observe several emergent patterns. Under symmetric noise, distilled samples consistently outperform full-data training beyond a critical noise threshold, even at 1 IPC. Under asymmetric noise, the distillation process tends to absorb structured label errors into the synthetic set, impairing generalization. For natural noise, dataset distillation remains robust at high noise levels but may degrade under mild noise due to overcompression of rare or hard examples.

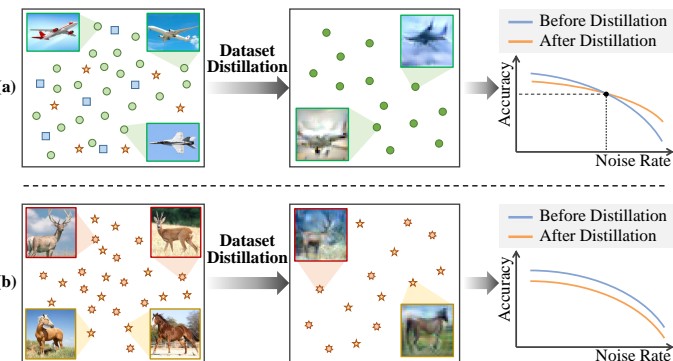

Figure 1: Dataset distillation under different types of label noise. (a) Under **symmetric noise**, where label corruption is random and unstructured, existing distillation methods effectively extract consistent patterns and act as implicit denoisers. (b) Under **asymmetric noise**, where corruption follows structured semantic confusion (e.g., visually similar classes), the distillation process may preserve these noisy patterns, leading to degraded generalization and entangled semantics in the distilled set.

These findings suggest that dataset distillation does not merely reflect memorization but rather performs task-aware semantic filtering—a perspective that guides our theoretical analysis. From a more general viewpoint, this observation motivates us to reinterpret dataset distillation as a process of semantic compression, one that selectively preserves information and suppresses noise.

To understand these phenomena, we propose a unified theoretical framework that quantifies the required IPC for generalization under noisy supervision. Our analysis incorporates the clean label proportion, data redundancy, and the intrinsic structure of label confusion. Under symmetric noise, we show that IPC scales with the inverse of $(1 - \tau)$, where $\tau$ is the noise rate. For asymmetric noise, we link the effective IPC bound to the effective confusion class-count that quantifies the number of distinguishable semantic modes. For natural noise, we introduce an entropy-based metric $\kappa$ to characterize the number of distinguishable semantic classes, and derive IPC bounds via PAC-Bayes McAllester (1999) and information-theoretic principles Shannon (1948).

**Contributions.** Our major contributions can be summarized as follows:

- We propose a new perspective for robust model training under label noise by reinterpreting dataset distillation as an implicit denoising mechanism. This perspective avoids error-prone noise estimation loops and supports efficient, privacy-preserving data processing.

- We conduct the comprehensive empirical evaluation of representative dataset distillation methods under symmetric, asymmetric, and natural noise settings, revealing key patterns that differentiate clean and noisy regimes.

- For the first time, we derive information-theoretic and PAC-Bayes bounds on the required IPC for successful distillation under various noise types, and introduce entropy-based metrics for semantic compression analysis. These results provide both diagnostic and predictive tools for robust dataset distillation.

- Our framework additionally enables principled data quality assessment via noise and redundancy estimation, and provides theoretical guarantees for noise estimation *in the wild* by aligning distilled and full-dataset generalization performance.

## 2 RELATED WORKS

### 2.1 DATASET DISTILLATION

Dataset distillation aims to synthesize a compact set of synthetic samples—typically a few images per class (IPC)—that can train models to match the performance of full-data training Wang et al. (2018a). Existing methods fall into three main categories: *Meta-learning* approaches treat distillation as a bi-level optimization problem, using validation feedback to update synthetic samples Wang et al. (2018a); Nguyen et al. (2021). *Parameter matching* methods align model updates between real and synthetic data, using gradient matching Zhao et al. (2020) or trajectory supervision Guo et al. (2023). *Distribution matching* minimizes statistical divergence between real and synthetic distributions, with techniques like MMD Zhao & Bilen (2023) and dual-view alignment Zhang et al. (2024). Our study builds upon representative methods—DATM Guo et al. (2023), DANCE Zhang et al. (2024), and RCIG Loo et al. (2023)—and systematically analyzes their robustness under various types of label noise. This offers new perspectives on the interplay between semantic condensation and noisy supervision.

### 2.2 LEARNING WITH NOISY LABELS

Noisy label learning aims to enhance model robustness under label corruption from annotation errors or automation Zhang & Sabuncu (2018); Li et al. (2020a). Existing methods fall into three major categories: *Noise modeling* estimates the corruption process using transition matrices Natarajan et al. (2013); Yu et al. (2018), instance-dependent estimators Cheng et al. (2020); Yang et al. (2022), or privileged information Wang et al. (2024). *Representation learning* leverages contrastive techniques to improve noise resilience, including noise-aware frameworks Ciortan et al. (2021), twin-branch contrastive models Huang et al. (2023), and selective learning from clean samples Li et al. (2022). *Training strategy adjustment* involves filtering Han et al. (2018), dynamic correction Li et al. (2023), and reweighting Shu et al. (2019); Zhou et al. (2024), often guided by curriculum learning Jiang et al. (2018) or noise-aware objectives Bae et al. (2024). In particular, approaches Arpit et al. (2017); Han et al. (2018); Yu et al. (2019); Li et al. (2020b); Han et al. (2020); Xia et al. (2020) such as memorization analysis highlight the tendency of deep networks to overfit noisy data, suggesting the need for methods that reduce the influence of noisy samples. This also motivates us to distill noisy dataset into a compact subset that still enables effective training.

## 3 DATASET DISTILLATION UNDER SYMMETRIC NOISE

**Discovery I.** *When the label noise ratio $\tau$ surpasses a critical threshold, mainstream dataset distillation methods consistently yield higher validation accuracy than training on the full noisy dataset, despite using significantly fewer distilled samples.*

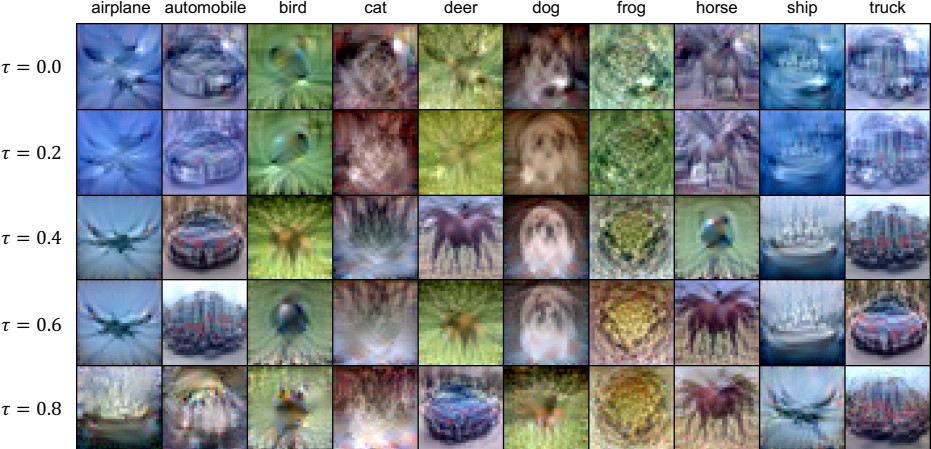

Figure 2: Images distilled by DATM on CIFAR-10 (IPC=1) under different symmetric noise.

To begin our analysis, we consider symmetric label noise, where each true label $y$ is randomly flipped to another class $y' \neq y$ with fixed probability $\tau$. Formally, the corrupted label $\tilde{y}$ follows:

$$p(\tilde{y} = y' \mid y) = \begin{cases} 1 - \tau, & \text{if } y' = y \\ \frac{\tau}{C-1}, & \text{if } y' \neq y \end{cases}$$

where $C$ denotes the number of classes. This noise model introduces class-agnostic corruption, making each instance equally likely to be mislabeled, and thereby serves as a testbed for evaluating the noise-robustness of distillation algorithms.

As shown in Fig. 3, dataset distillation consistently surpasses the noisy full-set baseline across all datasets when $\tau \geq 0.2$. In particular, at high noise levels (e.g., $\tau = 0.6, 0.8$), even a single distilled sample per class yields better generalization than training on the entire corrupted dataset. This indicates that distillation selectively retains clean patterns while suppressing noisy signals, acting as a strong implicit denoiser.

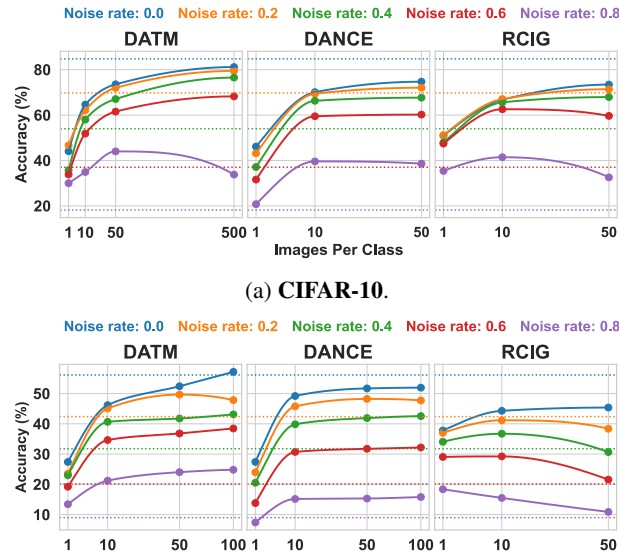

(a) **CIFAR-10**.

(b) **CIFAR-100**.

Figure 3: Distillation (Solid lines) *vs.* Full data (Dashed lines) under diverse **Symmetric** noise on CIFAR-10 and CIFAR-100.

✍ **Insight I.** These observations confirm that dataset distillation under symmetric noise serves as an effective denoising process. This supports the hypothesis that **the distillation process preferentially preserves common semantic structure across examples, while discarding outliers introduced by random label corruption**. Since symmetric noise is unstructured, its impact can be suppressed via semantic compression inherent in the distillation objective.

Nevertheless, a key question arises: *how many distilled images per class (IPC) are required to faithfully recover the task-relevant signal under a given noise level?* Addressing this question not only enables a more principled understanding of distillation dynamics, but also provides a means to estimate the underlying noise level from distillation behavior. This motivates a formal analysis of IPC bounds under symmetric noise, which we now develop in the following corollary.

**Corollary I ((Upper & Lower Bounds of IPC under Symmetric Noise and Redundancy)).** *Let $\tilde{S}$ be a noisy dataset with $C$ balanced classes, symmetric label noise rate $\tau \in [0, 1)$, and redundancy compression rate $r \in (0, 1]$. Denote by IPC the number of distilled images per class sufficient to preserve task-relevant semantics. Then the following holds (See Proof in Appendix):*

$$\frac{I_{\min}}{r \cdot (1 - \tau) \cdot I_{clean}} \leq \text{IPC} \leq \frac{|\tilde{S}| \cdot (1 - \tau) \cdot r}{C} \tag{1}$$

*Here, $I_{\min}$ is the minimum mutual information required per class for generalization, and $I_{clean}$ denotes the average information contribution from each clean, non-redundant sample. The upper bound is derived from a rate-distortion perspective, while the lower bound follows from PAC-Bayes and information bottleneck principles.*

## 4  DATASET DISTILLATION UNDER ASYMMETRIC NOISE

**Discovery II.** *Under asymmetric noise, dataset distillation methods tend to preserve structured label corruption patterns, resulting in synthetic samples that deviate from the true clean data distri-*

*bution. Even increasing the number of distilled images per class fails to fully recover the semantic diversity of the original clean dataset.*

To further understand the limitations of distillation, we evaluate its performance under **asymmetric noise**, a more realistic corruption pattern where label flips are class-dependent. Specifically, labels are more likely to be confused with semantically similar classes. This process is modeled by a conditional noise distribution $p(\tilde{y} = y' \mid y)$, where the flip probability $\tau(y \to y')$ depends on the semantic similarity between class $y$ and $y'$, such that:

$$p(\tilde{y} = y' \mid y) = \begin{cases} 1 - \sum_{y' \neq y} \tau(y \to y'), & \text{if } y' = y \\ \tau(y \to y'), & \text{if } y' \neq y \end{cases}$$

This setting captures structured confusion, e.g., TRUCK → AUTOMOBILE, which frequently occurs in human-labeled datasets. As shown in Fig. 4, most distillation methods fail to outperform training on the full noisy dataset, even at moderate noise levels ($\tau = 0.2, 0.4$). Only DATM achieves marginal improvements with sufficiently large IPC, indicating a partial robustness.

This behavior aligns with the intuition that dataset distillation compresses dominant patterns in the dataset, regardless of their correctness. When label corruption is structured, these patterns are no longer random outliers but form coherent—but incorrect—semantic clusters. Consequently, the distilled set not only retains clean signal but also encodes structured label transitions, which impairs its ability to represent the true clean distribution.

✍ **Insight II.** This observation highlights a fundamental limitation of current distillation methods: **common patterns captured during distillation are not necessarily clean**. When label noise is structured—such as in class-dependent or visually confounding cases—the distillation process may inadvertently preserve and even reinforce these incorrect seman-

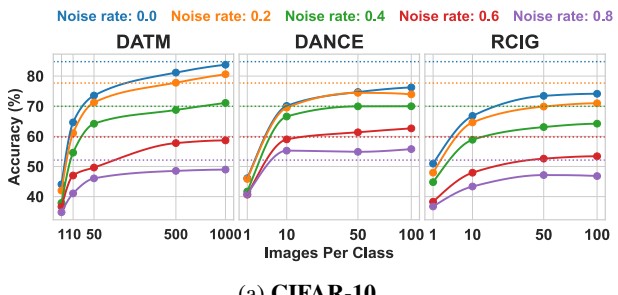

(a) **CIFAR-10**.

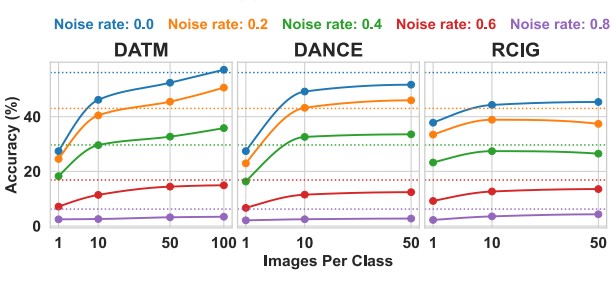

(b) **CIFAR-100**.

Figure 4: **Asymmetric** noise: Distillation lags full-data.

tics. Moreover, challenging but clean examples (e.g., tail classes or ambiguous instances) are likely to be underrepresented or compressed out, further degrading generalization.

These results call for a more nuanced treatment of dataset distillation in the presence of structured label noise. In particular, it becomes critical to understand: *how does the structure of asymmetric noise affect the semantic capacity of the distilled set, and what is the minimal IPC needed to recover meaningful signal?*

This motivates the following theoretical analysis, which characterizes the IPC bounds under asymmetric noise by considering the *effective confusion class-count* that quantifies the number of distinguishable semantic modes under asymmetric corruption (e.g., entropy-based, spectral effective rank, or MI-based).

**Corollary II ((Upper & Lower Bounds of IPC under Asymmetric Noise and Structured Redundancy)).** *Let $\tilde{\mathcal{S}}$ be a noisy dataset with $C$ balanced classes, asymmetric label noise transition matrix $T \in \mathbb{R}^{C \times C}$, and redundancy compression rate $r \in (0, 1]$. Define the average asymmetric noise rate as*

$$\tau := \frac{1}{C} \sum_{y=1}^{C} \sum_{y' \neq y} T_{y,y'} \in [0, 1),$$

*and define the* effective confusion class-count $C_{eff} \geq 1$ *as a functional of T that quantifies the number of distinguishable semantic modes under asymmetric corruption (e.g., entropy-based, spectral effective rank, or MI-based). Then the required number of distilled images per class (IPC) satisfies (See Proof in Appendix):*

$$\frac{I_{\min}}{r \cdot (1 - \tau) \cdot I_{clean}} \quad \leq \quad \text{IPC} \quad \leq \quad \frac{|\tilde{\mathcal{S}}| \cdot (1 - \tau) \cdot r}{C_{eff}} \tag{2}$$

*Here, $I_{\min}$ is the minimum mutual information required for generalization per class, and $I_{clean}$ denotes the average contribution from each clean, non-redundant sample. We will detail the formulation of $I_{\min}$, $I_{clean}$ and r in the Appendix. The lower bound follows from PAC-Bayes and information bottleneck theory, while the upper bound arises from rate-distortion principles under asymmetric, structured label corruption.*

## 5 DATASET DISTILLATION UNDER NATURAL NOISE

**Discovery III.** *Dataset distillation remains effective under real-world human-annotated label noise, with distilled models achieving competitive generalization at high noise rates, despite the absence of explicit noise modeling.*

To further validate the applicability of distillation in non-synthetic settings, we evaluate its performance under **natural label noise**, where labels are generated by real human annotators Wei et al. (2021). In contrast to synthetic noise models, natural noise is often class-dependent, ambiguous, and unstructured, making it difficult to model or correct. For CIFAR-10N, each sample is annotated by multiple humans; we consider three representative variants: **Random-$k$** ($\tau \approx 18\%$), where the $k$-th label is randomly selected; **Worst** ($\tau = 40.21\%$), where the most incorrect label is chosen; and **Aggre** ($\tau = 9.03\%$), where majority voting is used. CIFAR-100N follows a similar setup, with $\tau \approx 40.2\%$.

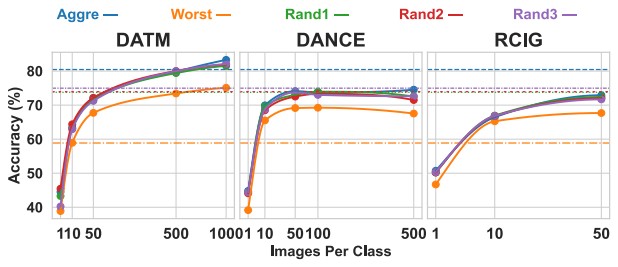

(a) **CIFAR-10N**.

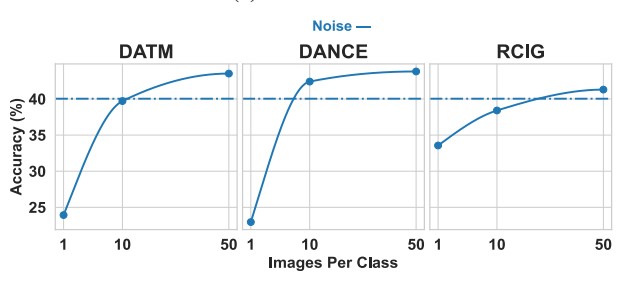

(b) **CIFAR-100N**.

Figure 5: CIFAR-10N/100N: Distillation is robust under high noise but may underperform at low noise (e.g., `Aggre`) due to overcompression.

Fig. 5 shows that distillation remains surprisingly robust in the `Worst` case: all evaluated methods (DATM, DANCE, RCIG) outperform the full noisy baseline with fewer than 10 IPC. However, under `Aggre`, where noise is minimal and majority labels are mostly correct, distillation performance deteriorates significantly.

This counterintuitive phenomenon arises from the interplay between sample scarcity and lossy compression. At low noise levels, the information bottleneck of distillation leads to excessive filtering of rare or ambiguous—but clean—samples, reducing the effective semantic coverage of the distilled set. In contrast, at higher noise rates, this compression selectively discards noisy labels, inadvertently improving signal fidelity.

✍ **Insight III.** These findings suggest that dataset distillation under natural noise behaves similarly to symmetric noise at high noise levels, functioning as a coarse denoiser by extracting robust semantic patterns. However, in low-noise settings, the limited IPC fails to preserve hard clean examples,

leading to suboptimal generalization. This reveals a key limitation of current distillation strategies: **they lack mechanisms to differentiate weak but useful signal from noise**, especially when noise is subtle or unstructured.

To address this, we seek a principled formulation of the required IPC under natural, uncontrolled label noise. Unlike synthetic or structured noise models, real-world noise lacks an explicit transition matrix or class-conditional distribution. Instead, we propose to model semantic confusability via the **label confusion matrix**, and define an entropy-based soft class cardinality $\kappa$ to quantify the effective number of distinguishable semantic modes.

This motivates the following theoretical analysis, which derives upper and lower bounds on IPC under natural noise using information-theoretic and PAC-Bayes perspectives (Corollary-III) and further offers probabilistic guarantees for estimating the underlying noise rate based on distilled generalization performance (Corollary-IV).

**Corollary III ((Upper & Lower Bounds of IPC under Natural Noise)).** *Let $\tilde{\mathcal{S}}$ be a real-world noisy dataset with $C$ annotated classes and total size $|\tilde{\mathcal{S}}|$. Let $r \in (0, 1]$ denote the redundancy compression rate. Suppose $\mathbf{M} \in \mathbb{R}^{C \times C}$ is the class confusion matrix computed from model predictions, and define the normalized row-wise distributions the average confusion entropy and the effective distinguishable class number :*

$$\mathbf{P}_{i,:} := \frac{\mathbf{M}_{i,:}}{\sum_j \mathbf{M}_{i,j}}, \quad i = 1, \dots, C; H_i = -\sum_{j=1}^{C} \mathbf{P}_{i,j} \log \mathbf{P}_{i,j}; H_{avg} = \frac{1}{C} \sum_{i=1}^{C} H_i; \kappa = \exp(H_{avg})$$

*The required number of distilled images per class (IPC) satisfies (See Proof in Appendix):*

$$\frac{I_{\min}}{\kappa \cdot r \cdot I_{clean}} \quad \leq \quad \text{IPC} \quad \leq \quad \frac{|\tilde{\mathcal{S}}| \cdot r}{\kappa} \tag{3}$$

*Here, $I_{\min}$ is the minimum mutual information required for generalization, and $I_{clean}$ denotes the average contribution per clean, de-redundified sample. The upper bound is derived from rate-distortion compression under class confusion; the lower bound follows from PAC-Bayes and information bottleneck perspectives. Observe that the construction of the confusion-derived $\kappa$ inherently accounts for average label correctness and semantic confusability, thus precluding the need for a separate $(1 - \tau)$ factor.*

**Corollary IV ((Heuristic PAC-Bayes Estimate of Noise Rate)).** *Let $\tilde{\mathcal{S}} \sim \tilde{\mathcal{D}}$ be a noisy dataset with $C$ balanced classes and total size $n = |\tilde{\mathcal{S}}|$, where the true (unknown) label noise rate is $\tau \in [0, 1]$. Suppose a distilled dataset $\mathcal{S}_d$ of size $m = C \cdot \text{IPC}$ achieves validation performance comparable to training on $\tilde{\mathcal{S}}$. Assume all examples in $\mathcal{S}_d$ are clean and informative. Then, as a* rule-of-thumb, *with probability at least $1 - \delta$ (See Proof in Appendix),*

$$1 - \frac{\alpha m}{n} \quad \leq \quad \tau \quad \leq \quad 1 - \frac{\alpha m}{n} \quad + \quad \sqrt{\frac{\text{KL}(Q \| P) + \log\left(\frac{2\sqrt{m}}{\delta}\right)}{2m}} \quad .$$

*Here, $Q$ is the posterior distribution over hypotheses trained on the distilled dataset $\mathcal{S}_d$, $P$ is the prior distribution (data-independent), and the bound is derived via the PAC-Bayes generalization framework. The parameter $\alpha$ is an information efficiency factor that quantifies how much clean information a distilled example is assumed to represent.*

# 6 EXPERIMENTS

This section follows a structured pipeline: we begin by outlining the experimental details, followed by a summary of key observations, and conclude with insightful conclusions.

## 6.1 IMPLEMENTATION DETAILS

**Dataset and Noise Construction.** We evaluate all distillation methods on three standard benchmarks: CIFAR-10, CIFAR-100 Krizhevsky et al. (2009), and Tiny ImageNet Le & Yang (2015). To

assess robustness under label corruption, we construct noisy variants using two canonical settings: *symmetric* and *asymmetric* noise, following the protocols in Patrini et al. (2017); Zhang & Sabuncu (2018). In the symmetric setting, labels are flipped uniformly at random to any of the remaining classes. For asymmetric noise, label transitions follow semantically coherent mappings: e.g., in CIFAR-10, TRUCK → AUTOMOBILE, BIRD → AIRPLANE, DEER → HORSE, and symmetric transitions between CAT ↔ DOG. In CIFAR-100, the 100 fine-grained categories are grouped into 20 superclasses, and label corruption occurs within each superclass via deterministic pairwise substitutions. Additionally, we adopt CIFAR-N Wei et al. (2021), which incorporates human-annotated noisy labels and better reflects natural noise conditions.

**Architecture and Evaluation Protocol.** Following prior work Zhao & Bilen (2021; 2023); Cazenavette et al. (2022), we employ lightweight ConvNet backbones for all experiments: a three-layer ConvNet for CIFAR datasets, and a four-layer variant for Tiny ImageNet. We evaluate the performance of each distilled dataset by training a fresh model from scratch on the distilled samples and reporting test accuracy on a held-out clean test set. For RCIG, we additionally apply the recommended data augmentation during training. All results are averaged across multiple seeds, and final test accuracy is reported at the end of the distillation stage.

## 6.2 FROM THEORY TO PRACTICE: HOW NOISE SHAPES DATASET DISTILLATION

Our theoretical analysis (Corollaries I–V) is validated across different noise models, confirming that dataset distillation functions as a semantic compressor which prioritizes consistent structures while filtering noise.

**Empirical Analysis under Symmetric Noise.** On CIFAR-10 and CIFAR-100, we validate the theory from two angles: varying IPC at fixed noise and varying noise at fixed IPC. With a moderate noise level (e.g., $\tau = 0.4$), very small IPC (e.g., 1) yields poor performance, but accuracy rises quickly with more samples and soon surpasses the full-data baseline, saturating once sufficient clean information is captured (Fig. 3a, 3b). Conversely, when IPC is fixed (e.g., 10) and noise increases, full-data training deteriorates rapidly while distilled sets remain stable, even outperforming full-data at high noise ($\tau = 0.6, 0.8$, Fig. 6). These results confirm **Corollary I**: under symmetric noise, IPC scales inversely with the clean-label proportion $(1 - \tau)$, and dataset distillation acts as an effective denoiser when corruption is severe.

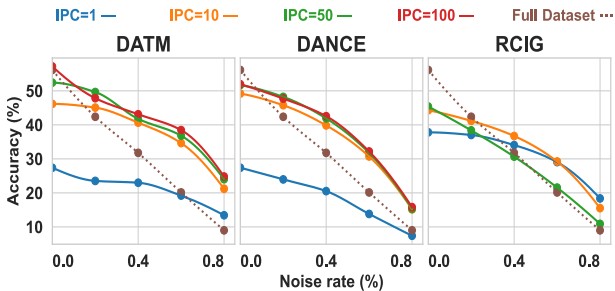

Figure 6: CIFAR100 results with symmetric noises.

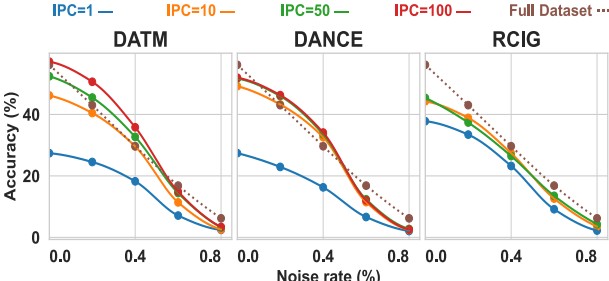

Figure 7: CIFAR100 results with asymmetric noises.

**Empirical Analysis under Asymmetric Noise** On CIFAR-10 and CIFAR-100, we find that increasing IPC under asymmetric noise (e.g., 40%) steadily improves performance but rarely surpasses the full-data baseline (Fig. 4a, 4b). Unlike the symmetric case, the gap persists because distilled sets inevitably absorb structured confusion among semantically similar classes, reducing the effective number of distinguishable modes. When IPC is fixed (e.g., 10) and the noise rate increases (Fig. 7), full-data accuracy drops sharply while distilled models degrade more slowly, yet never overtake full-data training, even at high noise. These results confirm that under asymmetric noise, IPC is governed not only by the clean-label proportion but also by the effective mode count $C_{\text{eff}}$, with structured corruption embedding spurious semantics that fundamentally limit the gains of distillation.

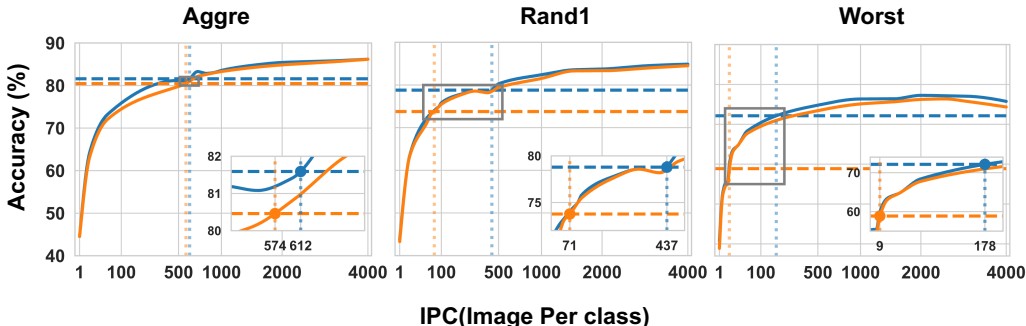

Figure 8: CIFAR10N results under different natural noises. Blue curves denote **Best Acc**, and yellow curves denote **Last Acc**.

**Empirical Analysis under Natural Noise**   Unlike symmetric or asymmetric noise, natural noise lacks an explicit transition matrix or controllable rate, making direct analysis of fixed IPC with varying noise infeasible. Instead, we use the entropy-based effective class cardinality $\kappa = \exp(H_{\mathrm{avg}})$ to proxy annotation quality. As shown in Fig. 5 and Fig. 8, increasing IPC under a fixed annotation protocol steadily improves performance by compensating for semantic loss, while fixed IPC across protocols reveals large performance gaps: high-$\kappa$ annotations (e.g., Worst) require more distilled samples but still benefit from implicit denoising, whereas low-$\kappa$ protocols (e.g., Aggre) induce over-compression, losing rare yet informative examples. Overall, these findings confirm that under natural noise, distillation robustness is driven not by explicit noise rate but by the structure of semantic confusion, with $\kappa$ serving as a practical surrogate for annotation quality and IPC demand.

**Conclusion**   Overall, these results consolidate our theoretical insights: (i) under symmetric noise, IPC scales inversely with the clean-label proportion $(1 - \tau)$ and distillation exhibits a unique denoising advantage; (ii) under asymmetric noise, generalization is constrained by $C_{\mathrm{eff}}$ and IPC scaling alone cannot offset structured corruption; (iii) under natural noise, robustness is governed by the structure of semantic confusion rather than explicit noise rate, with $\kappa$ serving as a practical surrogate for predicting IPC demand.

### 6.3    ACCURACY GAP AS A LABEL-FREE PROXY FOR DATA QUALITY

Finally, **Corollary-IV** (Fig. 8) provides a PAC-Bayes-based perspective on estimating the noise rate $\tau$ from the distilled sample size $m$. In particular, when distilled datasets of size $m = C \cdot IPC$ achieve comparable performance to full-data training, the lower bound $1 - \alpha m/n$ on $\tau$ becomes tighter as $m$ decreases. This means that a smaller distilled set either reflects high redundancy in the original dataset (where only a few samples are sufficient to represent the underlying semantics) or indicates the presence of substantial noise (where distillation successfully isolates a small set of clean, task-relevant examples). This interpretation aligns with the empirical ordering of annotation protocols, where $\kappa(\text{Aggre}) < \kappa(\text{Rand-1}) < \kappa(\text{Worst})$, reflecting increasing semantic confusion. When combined with the accuracy gap between distilled and full-data models, these observations provide a practical, label-free signal for unsupervised quality assessment: small gaps suggest redundancy or low noise, while large discrepancies reveal latent corruption. Together, these findings reinforce the view that dataset distillation is not merely a data reduction tool, but a principled mechanism of *semantic filtering* shaped by the structure of noise.

## 7    CONCLUSION AND LIMITATION

We show that dataset distillation serves as an implicit denoising mechanism by compressing semantic structures while filtering out noise. Through comprehensive experiments and PAC-Bayes-based analysis, we establish its robustness under symmetric and natural label noise, and quantify the required image-per-class (IPC) for effective generalization. However, our study is limited to class-balanced datasets. Extending this framework to long-tailed or imbalanced scenarios—where rare class compression may be more lossy—remains an important direction for future work.

ETHICS STATEMENT

This work does not involve human subjects, animal experiments, or sensitive personal data. The datasets used (e.g., CIFAR-10/100, Tiny ImageNet) are publicly available benchmark datasets commonly used in machine learning research and do not contain personally identifiable information. Our method focuses on synthetic data generation for label denoising and does not introduce new harmful applications. We have carefully reviewed the ICLR Code of Ethics and confirm that this submission complies with its principles regarding fairness, privacy, and research integrity. No potential conflicts of interest exist among the authors.

REPRODUCIBILITY STATEMENT

To ensure reproducibility, we provide the following resources: (1) All implementation details, including network architectures, hyperparameters, and training protocols, are described in Section 6.1 and the Appendix. (2) Theoretical derivations and assumptions for Corollaries I–IV are fully detailed in Appendix B.2→Appendix B.5. (3) Random seeds are fixed and reported; all results are averaged over multiple runs.

LLM USAGE STATEMENT

Large Language Models (LLMs) were used in this work solely as a general-purpose writing assistance tool—for example, to improve grammar, clarify phrasing, or check technical terminology in the manuscript. LLMs did not contribute to the conception of the research idea, theoretical analysis, experimental design, or interpretation of results. All scientific content, including equations, algorithms, and claims, was developed and verified by the authors. No LLM was used to generate novel technical content or to draft substantial portions of the paper. As required by ICLR policy, we confirm that LLMs are not listed as authors, and we take full responsibility for all content under our names.

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

APPENDIX

# A  ALL IMPLEMENTATION DETAILS.

We evaluate these benchmarks using datasets CIFAR-10/100 Krizhevsky et al. (2009), and tiny-ImageNet Le & Yang (2015), curating noisy versions Patrini et al. (2017); Zhang & Sabuncu (2018) using symmetric and asymmetric noises. Specifically, for asymmetric noise, labels are flipped to similar classes (e.g., in CIFAR-10: TRUCK → AUTOMOBILE, BIRD → AIRPLANE, DEER → HORSE, CAT ⟷ DOG; in CIFAR-100, the 100 classes are grouped into 20 superclasses, with each subclass flipping to the next within the same superclass). Additionally, we also adopt a more challenging version CIFAR-N Wei et al. (2021) that mimics human annotations. Following Zhao & Bilen (2021; 2023); Cazenavette et al. (2022), we employ a simple ConvNet Sagun et al. (2018) architecture for distillation: a three-layer ConvNet for CIFAR and a four-layer ConvNet for Tiny-ImageNet. Performance is evaluated based on test accuracy on distilled datasets, following the evaluation protocols of DATM, DANCE, and RCIG, with data augmentation applied for RCIG as recommended in the original work. Final test accuracies are reported throughout the distillation process unless otherwise stated.

## A.1  BENCHMARKING DATASET DISTILLATION METHODS

Dataset distillation has seen rapid development in recent years. To ensure coverage of core methodologies, we select three representative state-of-the-art approaches: parameter matching (DATM Guo et al. (2023)), distribution matching (DANCE Zhang et al. (2024)), and meta-learning (RCIG Loo et al. (2023)).

**Benchmark-I: DATM** aligns training trajectories between real and synthetic data by minimizing the discrepancy between model parameters obtained from both domains. This trajectory-based supervision allows the distilled data to inherit effective learning dynamics from real data. To address scalability limitations, a recent memory-efficient variant Cui et al. (2023) reduces unrolled gradient computation, enabling deployment on larger datasets.

**Benchmark-II: DANCE** extends distribution matching by addressing both intra-class compactness and inter-class separability. It fuses randomly initialized and expert-trained encoders to construct dual-view features, and introduces a calibration loss that encourages the expert model to better adapt to synthetic data, improving representation fidelity under domain shifts.

**Benchmark-III: RCIG** formulates dataset distillation as a bilevel optimization problem using meta-learning. It reparameterizes implicit gradients with neural tangent kernel (NTK) approximations, enabling analytical gradient computation. This framework jointly optimizes the distilled set and backbone parameters, offering theoretical insights and strong performance in low-data regimes.

**Remark.** These three methods capture the diversity of modern dataset distillation paradigms. We adopt them as benchmarks to evaluate performance under noisy labels, providing insights into their robustness and applicability for privacy-sensitive or label-imperfect settings.

# B  THEORETICAL DERIVATIONS AND ASSUMPTIONS FOR COROLLARIES I–IV

## B.1  SETUP AND NOTATION

Table 1 summarizes the core notation used throughout this work. In particular, IPC denotes the distilled images per class, which together with the number of classes $C$ determines the total distilled set size $m$. The parameters $\tau$ and $r$ capture noise severity and redundancy compression, respectively, while $I_{\min}$ and $I_{\text{clean}}$ characterize the information-theoretic requirements for generalization. For structured noise, $T$ denotes the transition matrix and its derived quantity $C_{\text{eff}}$ measures the effective number of semantic modes preserved. For natural noise, the entropy-based proxy $\kappa$ serves an analogous role, quantifying annotation quality via effective distinguishability. These symbols form the foundation of the theoretical corollaries and experimental analyses presented in this paper.

| Symbol | Definition |
|--------|------------|
| IPC | Distilled images per class; total distilled size $m = C \cdot \text{IPC}$. |
| $C$ | Number of classes (balanced unless stated). |
| $\tau$ | Average noise rate (symmetric: flip prob.; asymmetric: mean transition rate). |
| $r$ | Redundancy compression rate ($0 < r \leq 1$). |
| $I_{\min}$ | Minimum MI required per class for generalization. |
| $I_{\text{clean}}$ | Average MI per clean, non-redundant sample. |
| $T$ | Transition matrix for asymmetric noise, $T_{y,y'} = P(\tilde{y} = y' \mid y)$. |
| $C_{\text{eff}}$ | Effective confusion class count (entropy-based, effective rank, or MI). |
| $\kappa$ | Effective distinguishable class number under natural noise, $\kappa = \exp(H_{\text{avg}})$. |

Table 1: Notation used throughout the analysis.

## B.2 TIGHT PER-CLASS IPC BOUNDS UNDER SYMMETRIC NOISE

**Corollary I** (Symmetric Noise: Per-class IPC Bounds). *Let $\tilde{\mathcal{S}}$ be a dataset with $C$ balanced classes, total size $|\tilde{\mathcal{S}}|$, corrupted by* symmetric *label noise with rate $\tau \in [0, 1)$: $\mathbb{P}(\tilde{y} = y) = 1 - \tau$ and $\mathbb{P}(\tilde{y} = y' \neq y) = \tau/(C - 1)$. Let $r \in (0, 1]$ denote the redundancy compression rate (the fraction of unique, task-relevant samples after removing redundancy). Let $I_{\min}$ be the minimum mutual information required per class for generalization, and $I_{clean}$ the average information contribution of a clean, non-redundant sample. If $IPC$ denotes the number of distilled images* per class*, then*

$$\frac{I_{\min}}{r \cdot (1 - \tau) \cdot I_{clean}} \;\leq\; \text{IPC} \;\leq\; \frac{|\tilde{\mathcal{S}}| \cdot (1 - \tau) \cdot r}{C} \quad.$$

*Assumptions.*

- (A1) **Class balance.** $\tilde{\mathcal{S}}$ *has $C$ classes with equal prior $1/C$; samples are i.i.d.*

- (A2) **Symmetric noise.** *Labels are corrupted independently of inputs given the true label, with flip rate $\tau$ as above.*

- (A3) **Redundancy model.** *Among any per-class subset of size $n$, at most $rn$ samples are unique and task-relevant; the rest are redundant or near-duplicates.*

- (A4) **Per-sample information.** *Each clean, non-redundant sample contributes at most $I_{clean}$ mutual information about the per-class task, and noisy or redundant samples contribute no additional information (by the Data Processing Inequality and subadditivity).*

- (A5) **Per-class sufficiency.** *To achieve generalization on class $c$, the distilled set must carry at least $I_{\min}$ mutual information about class-$c$ decision-relevant factors.*

**Lemma 1 (Clean non-redundant budget).** Under (A1)–(A3), the maximum number of clean, non-redundant samples per class is

$$N_{\max}^{(c)} \;\leq\; r \cdot \frac{(1 - \tau) \cdot |\tilde{\mathcal{S}}|}{C}.$$

*Proof.* The expected number of clean samples per class is $\frac{(1-\tau) \cdot |\tilde{\mathcal{S}}|}{C}$. By (A3), at most a fraction $r$ of these are non-redundant.

**Lemma 2 (Information upper bound).** Under (A4), the information carried by a per-class subset of size $n^{(c)}$ is at most

$$I_{\text{tot}}^{(c)} \;\leq\; I_{\text{clean}} \cdot \min\{n^{(c)}, N_{\max}^{(c)}\}.$$

*Proof.* Noisy or redundant samples do not increase information; the total is upper bounded by the number of clean, non-redundant samples multiplied by $I_{\text{clean}}$.

**Upper bound.** Since distillation cannot create more information than is available, by Lemma 1 and 2 the maximum per-class distilled size is

$$\text{IPC} \ \leq \ N_{\max}^{(c)} = r \cdot \frac{(1-\tau) \cdot |\tilde{\mathcal{S}}|}{C}.$$

**Lower bound.** To satisfy (A5), we require $I_{\text{tot}}^{(c)} \geq I_{\min}$. By Lemma 2, in expectation

$$\mathbb{E}[I_{\text{tot}}^{(c)}] \ \leq \ I_{\text{clean}} \cdot \text{IPC} \cdot (1-\tau) \cdot r,$$

because each distilled sample contributes useful information with probability at most $(1-\tau) \cdot r$ (clean with probability $1-\tau$ and non-redundant with probability $r$). Thus, to guarantee $I_{\min}$ we must have

$$\text{IPC} \ \geq \ \frac{I_{\min}}{I_{\text{clean}} \cdot (1-\tau) \cdot r}.$$

**Conclusion.** Together the bounds yield

$$\frac{I_{\min}}{r \cdot (1-\tau) \cdot I_{\text{clean}}} \ \leq \ \text{IPC} \ \leq \ \frac{|\tilde{\mathcal{S}}| \cdot (1-\tau) \cdot r}{C}.$$

Both bounds are expressed in terms of *IPC*. The total distilled size is $m = C \cdot \text{IPC}$.

## B.3 Tight Per-class IPC Bounds under Asymmetric Noise

**Corollary II** (Asymmetric Noise: Per-class IPC Bounds with Effective Confusion). *Let $\tilde{\mathcal{S}}$ be a dataset with $C$ balanced classes and total size $|\tilde{\mathcal{S}}|$, corrupted by* asymmetric *label noise governed by a transition matrix $T \in \mathbb{R}^{C \times C}$. Let $\tau \in [0, 1)$ denote the average label-noise rate and $r \in (0, 1]$ the redundancy-compression rate (fraction of unique, task-relevant samples after removing redundancy). Let $I_{\min}$ be the minimal per-class mutual information required for generalization and $I_{clean}$ the average information contributed by a clean, non-redundant sample. Define the* effective confusion class-count $C_{eff} \geq 1$ *as a functional of $T$ that quantifies the number of distinguishable semantic modes under asymmetric corruption (e.g., entropy-based, spectral effective rank, or MI-based; see Assumption (A6) below). If $IPC$ denotes the number of distilled images* per class, *then*

$$\frac{I_{\min}}{r \cdot (1-\tau) \cdot I_{clean}} \ \leq \ \text{IPC} \ \leq \ \frac{|\tilde{\mathcal{S}}| \cdot (1-\tau) \cdot r}{C_{eff}} \quad .$$

*Assumptions.*

*(A1)* *Class balance. Classes are balanced with prior $1/C$; samples are i.i.d.*

*(A2)* *Asymmetric noise. Given the true label $y$, the observed label $\tilde{y}$ is drawn from row $T_{y,:}$ of a transition matrix $T$, independently of the input conditioned on $y$. The average noise rate is $\tau := \frac{1}{C} \sum_y \sum_{y' \neq y} T_{y,y'}$.*

*(A3)* *Redundancy model. Among any per-class subset of size $n$, at most $rn$ samples are unique and task-relevant; the rest are redundant/near-duplicate for the task.*

*(A4)* *Per-sample information. Each clean, non-redundant sample contributes at most $I_{clean}$ mutual information about per-class decision-relevant factors; noisy or redundant samples do not increase this amount (Data Processing Inequality + subadditivity).*

*(A5)* *Per-class sufficiency. To generalize on class $c$, the distilled per-class set must carry at least $I_{\min}$ mutual information about class-$c$ decision-relevant factors.*

*(A6)* *Effective confusion. The asymmetric noise induces a collapse of semantic distinctions into $C_{eff} \geq 1$ effective modes. Formally, $C_{eff}$ is any choice of functional that monotonically decreases as structured confusion strengthens and equals $C$ in the no-confusion limit. Typical estimators include:*

- Entropy-based: *Form the row-normalized matrices $P_{y,:} = T_{y,:}$; define $H_y = -\sum_{y'} P_{y,y'} \log P_{y,y'}$, $H_{avg} = \frac{1}{C} \sum_y H_y$, and $C_{eff} = \exp(H_{avg})$.*
- Spectral effective rank: *Let $\{\sigma_k\}$ be singular values of $T$, normalized by $S = \sum_k \sigma_k$, $p_k = \sigma_k/S$; set $C_{eff} = \exp(-\sum_k p_k \log p_k)$.*
- Mutual-information based: $C_{eff} = \exp(I(Y; \tilde{Y}))$ *with a fixed log base.*

*All these estimators are equivalent up to monotonic transformations and yield the same asymptotic behavior of IPC bounds.*

**Notation.** Write $\tilde{\mathcal{S}} = \mathcal{S}_{\text{clean}} \cup \mathcal{S}_{\text{noisy}}$, with expected clean mass $|\mathcal{S}_{\text{clean}}| = (1 - \tau) \cdot |\tilde{\mathcal{S}}|$. Let $|\mathcal{S}_{\text{clean}}^{\text{tot}}| = (1 - \tau) \cdot |\tilde{\mathcal{S}}|$ denote the (expected) total number of clean samples across all classes, and recall that IPC is *per-class*.

**Lemma 3 (Global clean non-redundant budget).** Under (A2)–(A3), the (expected) total number of clean, non-redundant samples is bounded by

$$N_{\max}^{\text{tot}} \leq r \cdot |\mathcal{S}_{\text{clean}}^{\text{tot}}| = r \cdot (1 - \tau) \cdot |\tilde{\mathcal{S}}|.$$

*Proof.* At most a fraction $r$ of all (expected) clean samples are non-redundant by (A3).

**Lemma 4 (Effective-mode allocation).** Under (A2) and (A6), asymmetric corruption collapses semantics into $C_{\text{eff}}$ effective modes. Any distilled set that preserves all distinguishable modes cannot allocate, on average, more than

$$\frac{N_{\max}^{\text{tot}}}{C_{\text{eff}}}$$

clean, non-redundant samples *per effective mode*.

*Proof.* By definition, $C_{\text{eff}}$ upper bounds the number of distinguishable semantic modes that survive confusion. Hence the total clean, non-redundant budget $N_{\max}^{\text{tot}}$ must be distributed across at least $C_{\text{eff}}$ modes.

**Upper bound.** Distillation cannot produce more clean, non-redundant information than available. By Lemma 3, the global budget is $N_{\max}^{\text{tot}} = r(1 - \tau)|\tilde{\mathcal{S}}|$. By Lemma 4, this yields at most $N_{\max}^{\text{tot}}/C_{\text{eff}}$ clean, non-redundant samples *per effective mode*. Since each original class must be represented within these effective modes and IPC counts per-class samples, it follows that

$$\text{IPC} \leq \frac{N_{\max}^{\text{tot}}}{C_{\text{eff}}} = \frac{|\tilde{\mathcal{S}}| \cdot (1 - \tau) \cdot r}{C_{\text{eff}}}.$$

**Lower bound.** As in the symmetric case, to satisfy per-class sufficiency (A5) we require $I_{\text{tot}}^{(c)} \geq I_{\min}$. By (A4), the per-sample information of a distilled item is upper bounded by $I_{\text{clean}}$ and is realized only when the item is clean and non-redundant. Without oracle access to cleanliness or non-redundancy, the maximal per-item probability of being both is $(1 - \tau)r$, hence

$$\mathbb{E}[I_{\text{tot}}^{(c)}] \leq I_{\text{clean}} \cdot IPC \cdot (1 - \tau) \cdot r \quad \Rightarrow \quad \text{IPC} \geq \frac{I_{\min}}{I_{\text{clean}} \cdot (1 - \tau) \cdot r}.$$

**Conclusion.** Combining the bounds gives

$$\frac{I_{\min}}{r \cdot (1 - \tau) \cdot I_{\text{clean}}} \leq \text{IPC} \leq \frac{|\tilde{\mathcal{S}}| \cdot (1 - \tau) \cdot r}{C_{\text{eff}}}.$$

Both statements are in *per-class* units. The total distilled size is $m = C \cdot \text{IPC}$.

### B.4 EFFECTIVE IPC BOUNDS UNDER NATURAL NOISE

**Corollary III (Natural Noise: Per-class IPC Bounds with Confusion-based Effective Classes).** *Let $\tilde{\mathcal{S}}$ be a dataset with $C$ annotated classes and total size $|\tilde{\mathcal{S}}|$, subject to* natural *human annotation*

*noise (uncontrolled, class-dependent, possibly ambiguous). Let $r \in (0, 1]$ denote the redundancy-compression rate (fraction of unique, task-relevant samples after removing redundancy). Let $I_{\min}$ be the minimal per-class mutual information required for generalization and $I_{clean}$ the average information contributed by a clean, non-redundant sample. Let $\kappa \geq 1$ denote the* effective number of distinguishable semantic modes *under natural noise, estimated from an empirical confusion matrix (cf. Assumption (A6) below). If $IPC$ denotes the number of distilled images* per class*, then*

$$\frac{I_{\min}}{\kappa \cdot r \cdot I_{clean}} \quad \leq \quad \text{IPC} \quad \leq \quad \frac{|\tilde{\mathcal{S}}| \cdot r}{\kappa} \quad .$$

***Assumptions.***

(A1) **Class balance.** *Classes are balanced with prior $1/C$; samples are i.i.d. unless stated.*

(A2) **Natural noise.** *Labels are noisy due to human annotation without a known parametric transition model; corruption is potentially class-dependent and unstructured.*

(A3) **Redundancy model.** *Among any subset of size $n$, at most $rn$ samples are unique and task-relevant; the rest are redundant/near-duplicate for the task.*

(A4) **Per-sample information.** *Each clean, non-redundant sample contributes at most $I_{clean}$ mutual information about per-class decision-relevant factors; noisy or redundant samples do not increase this amount (Data Processing Inequality + subadditivity).*

(A5) **Per-class sufficiency.** *To generalize on class $c$, the distilled per-class set must carry at least $I_{\min}$ mutual information about class-$c$ decision-relevant factors.*

(A6) **Confusion-based $\kappa$.** *Let $\mathbf{M} \in \mathbb{R}^{C \times C}$ be a class confusion matrix obtained from a fixed reference model (e.g., a cleanly trained teacher) evaluated on a clean validation split or via a controlled protocol. Let $\mathbf{P}_{i,:} = \mathbf{M}_{i,:}/\sum_j \mathbf{M}_{i,j}$ be row-normalized distributions, $H_i = -\sum_j \mathbf{P}_{i,j} \log \mathbf{P}_{i,j}$, and $H_{avg} = \frac{1}{C}\sum_i H_i$. Define $\kappa = \exp(H_{avg})$ (perplexity of the average row entropy), which absorbs both average label correctness and semantic confusability: greater confusion/noise $\Rightarrow$ larger $H_{avg} \Rightarrow$ larger $\kappa$.*

(A7) **Per-class allocation via effective modes.** *Each original class must be represented within the surviving $\kappa$ effective modes. Thus any global budget of clean, non-redundant samples is, on average, distributed across at least $\kappa$ distinguishable modes, which upper-bounds what any single class can effectively retain.*

**Lemma 5 (Global non-redundant budget).** Under (A3), the (expected) total number of non-redundant, task-relevant samples available from $\tilde{\mathcal{S}}$ is

$$N_{\max}^{\text{tot}} \leq r \cdot |\tilde{\mathcal{S}}|.$$

*Proof.* By (A3), at most a fraction $r$ of any pool is non-redundant; applying to the full dataset gives the bound.

**Lemma 6 (Mode-wise allocation).** Under (A6)–(A7), the global non-redundant budget is, in expectation, spread across at least $\kappa$ effective modes, hence the *per-mode* budget is at most

$$\frac{N_{\max}^{\text{tot}}}{\kappa}.$$

*Proof.* By definition, $\kappa$ counts the number of distinguishable semantic modes that survive natural confusion; thus the total usable budget cannot exceed an equal-split upper bound across these modes.

**Upper bound.** Distillation cannot create more clean, non-redundant information than available. By Lemma 5 and 6, the effective per-class distilled size (since each class must be captured within the surviving modes by (A7)) is bounded by

$$\text{IPC} \leq \frac{N_{\max}^{\text{tot}}}{\kappa} = \frac{|\tilde{\mathcal{S}}| \cdot r}{\kappa}.$$

**Lower bound.** To satisfy per-class sufficiency (A5), the distilled subset for class $c$ must provide at least $I_{\min}$ mutual information. By (A4), each distilled item contributes at most $I_{\text{clean}}$ if it is clean and non-redundant. Under natural confusion summarized by $\kappa$, the effective "useful" fraction per item is upper-bounded in expectation by $r/\kappa$ (redundancy $r$ and a $1/\kappa$ dilution across effective modes). Hence

$$\mathbb{E}[I_{\text{tot}}^{(c)}] \;\leq\; I_{\text{clean}} \cdot \text{IPC} \cdot \frac{r}{\kappa} \quad \Rightarrow \quad \text{IPC} \;\geq\; \frac{I_{\min}}{\kappa \cdot r \cdot I_{\text{clean}}}.$$

**Conclusion.** Combining the two bounds yields

$$\frac{I_{\min}}{\kappa \cdot r \cdot I_{\text{clean}}} \;\leq\; \text{IPC} \;\leq\; \frac{|\tilde{\mathcal{S}}| \cdot r}{\kappa},$$

in *per-class* units. No separate $(1 - \tau)$ factor appears because the confusion-derived $\kappa$ *absorbs* average label correctness and semantic confusability by construction; see (A6).

### B.5 PAC-BAYES HEURISTIC BOUNDS ON THE LABEL NOISE RATE

**Corollary IV** (Heuristic PAC-Bayes Estimate of the Noise Rate). *Let $\tilde{\mathcal{S}}$ be a dataset with $C$ balanced classes and total size $n = |\tilde{\mathcal{S}}|$, with unknown label-noise rate $\tau \in [0, 1]$. Let $\mathcal{S}_d$ be a distilled dataset of size $m = C \cdot IPC$ whose validation performance is comparable to training on $\tilde{\mathcal{S}}$. Assume that all distilled items are clean and informative.*

*Assumptions.*

*(A1)* ***Bounded loss & i.i.d.*** *The loss is bounded in $[0, 1]$ and the sample used for the empirical term is i.i.d.*

*(A2)* ***Prior/posterior.*** *A data-independent prior $P$ and a posterior $Q$ depending only on $\mathcal{S}_d$ (or the sample used in the empirical term) are fixed.*

*(A3)* ***Clean-content efficiency.*** *There exists $\alpha \in (0, 1]$ such that the clean content in $\tilde{\mathcal{S}}$ satisfies $n(1 - \tau) \geq \alpha m$; $\alpha = 1$ corresponds to "one distilled item carries at most the clean information of one clean example".*

*Then, as a* rule-of-thumb*, with probability at least $1 - \delta$,*

$$1 - \frac{\alpha m}{n} \;\leq\; \tau \;\leq\; 1 - \frac{\alpha m}{n} + \sqrt{\frac{\text{KL}(Q\|P) + \log\!\left(\frac{2\sqrt{m}}{\delta}\right)}{2m}} \quad .$$

**Proof.** *Lower bound.* From (A3), $n(1 - \tau) \geq \alpha m \Rightarrow \tau \geq 1 - \alpha m/n$.

*Upper bound.* By a McAllester-style PAC-Bayes inequality (bounded loss, prior $P$, posterior $Q$), with probability $\geq 1 - \delta$,

$$\mathbb{E}_{h \sim Q}[L(h)] \;\leq\; \mathbb{E}_{h \sim Q}\big[\hat{L}(h, \mathcal{S}_d)\big] \;+\; \sqrt{\frac{\text{KL}(Q\|P) + \log\!\left(\frac{2\sqrt{m}}{\delta}\right)}{2m}} \;=:\; \mathbb{E}_{h \sim Q}\big[\hat{L}(h, \mathcal{S}_d)\big] + \varepsilon_m.$$

To tolerate this generalization gap when matching full-data performance, relax (A3) to $n(1 - \tau) \geq \alpha m - n\varepsilon_m$, which yields $\tau \leq 1 - \alpha m/n + \varepsilon_m$. Combining the bounds gives the stated interval. The gap between lower and upper bounds is governed entirely by $\varepsilon_m$, which vanishes at rate $\mathcal{O}(1/\sqrt{m})$.

**Remarks.** (1) This statement is explicitly *heuristic*: $\mathcal{S}_d$ is not an i.i.d. draw from the clean distribution, so PAC-Bayes is used to calibrate a tolerable gap, not to certify a formal estimator of $\tau$. (2) If validation, not $\mathcal{S}_d$, is used for the empirical risk, replace $m$ above by the validation size $n_{\text{val}}$ and state that $Q$ depends only on that validation sample. (3) Setting $\alpha = 1$ yields a conservative interval; $\alpha < 1$ allows distilled samples to be more information-dense than individual clean examples. (4) Throughout the paper, IPC denotes *per-class* images and the total distilled size is $m = C \cdot \text{IPC}$, consistent with Corollaries I–III.

## B.6 ESTIMATION OF THEORETICAL PARAMETERS

To bridge the gap between theory and practice, we provide concrete strategies to estimate the abstract quantities appearing in our IPC bounds.

### B.6.1 $I_{\min}$ – MINIMUM MUTUAL INFORMATION REQUIREMENT

***Definition.*** The minimum amount of task-relevant mutual information per class required to ensure generalization. $I_{\min}$ can be estimated by the following strategies:

(1) **Few-shot thresholding.** Gradually increase IPC on a clean validation set and identify the smallest IPC at which test accuracy rises significantly above a random or majority baseline. The corresponding information content can be regarded as $I_{\min}$.

(2) **PAC-Bayes perspective.** Approximate $I_{\min}$ by the minimum mutual information necessary to achieve a bounded generalization gap or posterior variance under a PAC-Bayes bound.

(3) **Information bottleneck proxy.** Use the difference $H(z) - H(z|y)$ of latent representations to approximate the point where non-trivial information is retained.

### B.6.2 $I_{\text{CLEAN}}$ – INFORMATION CONTRIBUTION OF A CLEAN SAMPLE

***Definition.*** The average mutual information provided by a non-redundant, correctly labeled sample. $I_{\text{clean}}$ can be estimated by the following strategies:

(1) **Contrastive mutual information.** Employ InfoNCE or MINE estimators to approximate $I(x; y)$ on clean samples and average across the set.

(2) **Teacher model proxy.** Estimate as $I_{\text{clean}} \approx \mathbb{E}[H(p(y)) - H(p(y|x))]$ where the entropy gap of teacher predictions reflects per-sample informativeness.

(3) **Gradient-based proxy.** Measure the average gradient norm of clean samples during distillation; larger norms indicate higher information contribution.

### B.6.3 $r$ – REDUNDANCY COMPRESSION RATE

***Definition.*** The proportion of unique, non-redundant task-relevant samples after redundancy removal. $r$ can be estimated by the following strategies:

(1) **Embedding clustering.** Cluster sample embeddings (e.g., cosine similarity) and compute $r \approx \frac{\#\text{clusters}}{\#\text{samples}}$.

(2) **Effective rank.** Use the normalized singular value spectrum of the embedding covariance matrix: $r \approx \frac{\exp(H(p))}{n}$, $p_i = \frac{\sigma_i}{\sum_j \sigma_j}$ where $H(p)$ is the entropy of the singular value distribution.

(3) **Influence functions.** Let $\{IF(z_i; \theta)\}_{i=1}^n$ denote the influence function contributions of samples $z_i \in \mathcal{S}$ with respect to model parameters $\theta$. The *global redundancy* $r$ of the dataset $\mathcal{S}$ is defined as $r \approx \frac{\left\| \sum_{i=1}^n IF(z_i;\theta) \right\|_2}{\sum_{i=1}^n \|IF(z_i;\theta)\|_2}$. This ratio measures the alignment of individual influence vectors: $r \to 1$ indicates low redundancy where samples contribute coherently, while $r \to 0$ corresponds to high redundancy where contributions largely cancel out. In practice, $IF(z_i; \theta)$ can be approximated via Hessian-free methods, random projections, or low-rank spectral estimation.

## C MORE RESULTS.

We further validate our theoretical predictions on the more challenging Tiny-ImageNet dataset, which contains substantially more classes and greater semantic diversity than CIFAR benchmarks. As shown in Fig. 9, under *symmetric noise*, increasing IPC steadily improves performance, and at

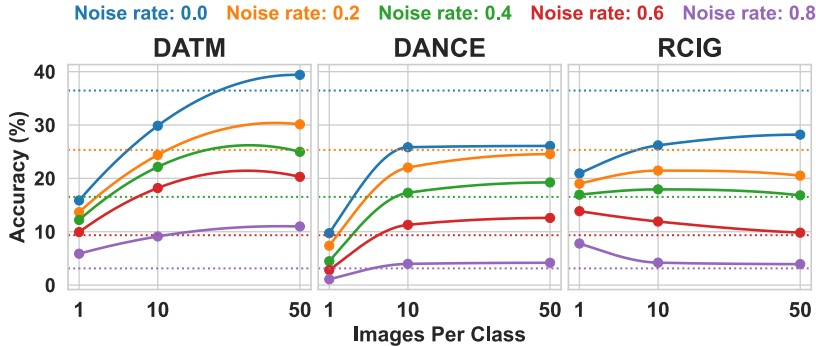

Figure 9: Tiny-ImageNet results with symmetric noises.

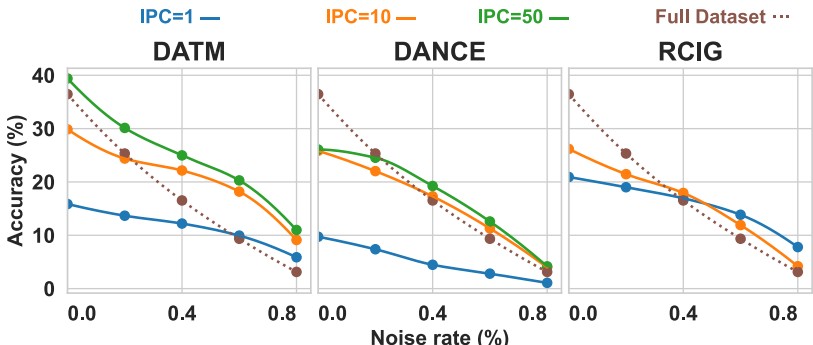

Figure 10: Tiny-ImageNet results with asymmetric noises.

high noise rates distilled models surpass full-data baselines. This again confirms that dataset distillation acts as a semantic compressor that filters unstructured corruption, consistent with Corollary I. In contrast, under *asymmetric noise* (Fig. 10), distilled models exhibit pronounced degradation: although larger IPC partially mitigates performance loss, the gap with full-data training remains significant. This observation supports Corollary II, as structured confusion among semantically similar classes reduces the effective number of distinguishable modes, causing distillation to inadvertently preserve spurious semantics. Overall, Tiny-ImageNet highlights that the denoising benefit of distillation holds primarily for random corruption, whereas structured label noise imposes a fundamental limit on generalization.

