# OpenReview forum: "Dataset Distillers Are Good Label Denoisers In the Wild"
_ICLR.cc/2026/Conference — Submitted to ICLR 2026_

### Official Review · Reviewer_upv1 · 2025-10-24

**Soundness:** 2
**Presentation:** 3
**Contribution:** 2
**Rating:** 2
**Confidence:** 4

**Summary:**

This paper investigates dataset distillation methods as implicit label denoisers, demonstrating that distillation can outperform full-data training under certain noise conditions. The authors provide both empirical evidence across CIFAR-10/100 and Tiny ImageNet, plus theoretical bounds for required images per class (IPC) under different noise types.

**Strengths:**

- The paper perceives dataset distillation as a denoising mechanism in order to further understand as to why distillation works.
- Tests three distinct noise types (symmetric, asymmetric, natural) provides a thorough analysis.
- PAC-Bayes bounds add rigor beyond purely empirical observations and probably are useful in real-world training cases.

**Weaknesses:**

This analysis in this paper offers a great insightful perspectives and lays a strong foundation that helps to further probe along the following lines and even add more value to the broader community. On that note, the paper does not investigate any of the following useful perspectives.

- The experiments use basic 3-4 layer ConvNets. How do these findings translate to modern architectures like ResNets, Vision Transformers, Diffusion models or other state-of-the-art methods?

- What about real-world dataset scales? The analysis is  carried out only on CIFAR and Tiny ImageNet raises serious scalability questions. Would these patterns hold on ImageNet, or larger datasets with hundreds of classes, for that matter datasets with high class imbalance?

- On that note, why assume class balance? Real-world datasets are often imbalanced, how does class imbalance affect the denoising properties? Are the proposed PAC-Bayes bounds generalize to imbalanced cases as well? If so, a demonstration through an example benchmark like ImageNet would greatly improve the quality of the paper.

- There is a clear finding that distillation fails with asymmetric noise but that finding is not further explored. This actually can be considered as a separate problem and a dedicated solution. This seems like a critical limitation that undermines the practical usability.

- Where are the dedicated denoising baselines? You compare against full-data training but not against established label denoising methods like DivideMix, Co-teaching, or meta-learning approaches. How does distillation compare to methods explicitly designed for noisy labels?

- What about data augmentation? Strong augmentation strategies can also act as implicit denoisers. How does distillation compare to simply using better augmentation on the full noisy dataset?

- There is no analysis on how the distilled samples would appear. Note that Deep nets are notoriously good at producing noisy images that can be deceived as real in-distribution samples. Therefore, it is highly recommended to check whether the distilled images semantically meaningful or not? Are they similar to the original clean samples or not? Do they preserve important visual features or just exploit model biases?

- Will the distilled samples from one architecture generalize to train different architectures? How long the change in architectures take to learn or produce decent results?

- What are the cost implications of doing distillation followed by training? Meaning, Is it computationally expensive to train data distillers followed by training on the end task or training on full noisy dataset?

**Questions:**

Please refer to the above weaknesses section for questions.

---

### Official Review · Reviewer_2b5n · 2025-10-31

**Soundness:** 3
**Presentation:** 3
**Contribution:** 2
**Rating:** 4
**Confidence:** 4

**Summary:**

This paper systematically studies dataset distillation under various labeled noise conditions, e.g., symmetric noise, asymmetric noise, and natural noise. Further, this paper reveals a counterintuitive yet insightful and interesting phenomenon: when exceeding the critical noise ratio, the distilled dataset can outperform the full-data training. The paper is also provided  empirical evidence on CIFAR-10, CIFAR-100, Tiny-ImageNet, etc, and covers the mainstream dataset distillation methods, e.g., DATM, DANCE, RCIG. Furthermore, the paper offers a theoretical explanation based on information theory and PAC-Bayes analysis, deriving upper and lower bounds on the required IPC for effective learning under different noise regimes.

**Strengths:**

1. The paper reveals a counterintuitive but highly interesting and insightful phenomenon, which broaden the potential application of dataset distillation as a denoising or robustness-enhancing technique for noisy dataset.
2. The paper offers comprehensive theoretical analysis based on information theory and PAC-Bayes, providing the upper and lower bounds on the required IPC, to support and explain the observed phenomena.
3. The presentation is clear with figures and equations.

**Weaknesses:**

1. [major] Although the observed phenomenon is interesting, its applicability appears to be rather limited and dependent on the dataset and adopted dataset distillation method. Specifically, under symmetric noise condition, the performance of the distilled dataset begins to outperform training on the full dataset when the noise ratio reaches around 20%. However, for more realistic asymmetric noise, the threshold rises above 40% on the CIFAR-10 dataset (which is quite extreme in real-world scenarios). This trend becomes inconsistent when using more complex datasets (like CIFAR-100), the advantage only appears in the 20% to 40% noise range and disappears at higher noise ratios. Furthermore, this phenomenon is not observed at all for RCIG. These results suggest that while this finding is conceptually interesting, it may be difficult to generalize to different datasets or noise settings, making it challenging to draw unified conclusions.
2. [minor] This experimental verification focuses primarily on small-scale image classification benchmarks, e.g. CIFAR-10, CIFAR-100, and Tiny-ImageNet, and lacks analysis of larger datasets.
3. [minor] This paper demonstrates that dataset distillation has an implicit denoising effect in high-noise condition, but it does not directly compare it with existing noise-resistant training or label correction methods, e.g., Co-teaching [1], DivideMix[2], GCELoss[3].

[1] Co-teaching: Robust Training of Deep Neural Networks with Extremely Noisy Labels

[2] DivideMix: Learning with Noisy Labels as Semi-supervised Learning

[3] Generalized Cross Entropy Loss for Training Deep Neural Networks with Noisy Labels

**Questions:**

1. The theoretical analysis in the paper is based on idealized assumptions such as independent noise, but these assumptions may not apply to real-world scenarios. For example, in Figure 4(b) with asymmetric noise, under the setting of CIFAR-100 and RCIG method, this phenomenon does not appear. Could the authors explain whether these theoretical results still apply to more complex and realistic situations? How do we select the IPC and determine if this phenomenon exists?
2. Could the authors clarify whether the observed phenomenon is expected to generalize to larger or more complex datasets?
3. The paper demonstrates that dataset distillation has an implicit denoising effect in high-noise condition, but it lacks comparison with existing noise-robust training such as Co-teaching, DivideMix, and GCELoss.

---

### Official Review · Reviewer_cF8n · 2025-11-01

**Soundness:** 2
**Presentation:** 2
**Contribution:** 2
**Rating:** 2
**Confidence:** 4

**Summary:**

The paper study the performance of existing dataset distillation methods (such as DM, DANCE, and RCIG) in noisy label scenarios. The authors claim that dataset distillation itself can serve as a potential denoising mechanism. They find that dataset distillation methods excel at filtering out symmetric noise and is robust to high natural noise, but struggle with asymmetric noise. They also use information theory and PAC-Bayes theory to propose a relationship (upper or lower bound) between the number of samples required for data distillation and the noise rate.

**Strengths:**

1. This paper reinterprets dataset distillation as an implicit denoising process through semantic compression.
2. They use information-theoretic and PAC-Bayes to give bounds for dataset distillation under different noise regimes
3. Well-designed small-scale experiments are provided to show the denoising effectiveness of distillation under high symmetric noise and also reveal its failure under asymmetric noise.

**Weaknesses:**

1. The theoretical bounds derived in Corollaries I-III need stronger empirical validation. Currently, the experimental results only show qualitative consistency with the theoretical trends. However, since this paper focuses on the analysis of new insight and findings from observational results, rather than proposing new solutions, qualitative analysis is insufficient. Experiments need to be designed to quantitatively test these IPC boundaries, and to discuss the gap between the theoretical boundaries and the practically achievable limits, as well as the possible causes of this gap. Is the reason why performance shown in Figure 4 and 5 cannot be improved because this theoretical limit has been reached?
2. In Corollary-II，authors try to characterize the IPC bounds under asymmetric noise by considering the rank of the noise transition matrix and its impact on semantic diversity. How is semantic diversity manifested? Could you explain in detail the relationship between semantic diversity and boundary?
3. In the formulas of conditional noise distribution shows in Discovery II, how is the flip probability specifically implemented based on semantic similarity? Please give a detailed explanation.
4. In Figure 5, authors gives that CIFAR-100N follows a similar setup, but only “Noise” setting is shown in the figure. Please add any other settings that are the same as those for the CIFAR-10N, or explain why you don’t use those settings.
5. In the section of “From Theory to Practice: How Noise Shapes Dataset Distillation”, only general and macro-level analysis is given, such as IPC scales inversely with the clean label proportion, which has already been claimed in the
previous section. Furthermore, there is a significant amount of repetition between Figures 5 and 7.
6. Further validation on more datasets and more distillation models is needed to demonstrate that the findings proposed in the paper are universally applicable to distillation methods and image label noise problems, rather than being limited to the 3 distillation models mentioned in the paper and the CIFAR dataset.

**Questions:**

Please refer to weaknesses.

---

### Official Review · Reviewer_9Gpj · 2025-11-09

**Soundness:** 3
**Presentation:** 3
**Contribution:** 3
**Rating:** 4
**Confidence:** 3

**Summary:**

This paper studied dataset distillation in the noise label setting. Empirical experiments show that data distillation shows improvement on results under symmetric noise and natural noise but decreased performance on asymmetric noise. Also the paper studied relationship between accuracy and IPC under symmetric and asymmetric conditions both empirically and theoretically.

**Strengths:**

The paper is well organized and presented.
The experiments is thorough under symmetric, asymmetric  and natural noise, with different noise level and IPC, providing promising results for analysis

**Weaknesses:**

See questions below

**Questions:**

The theoretical analysis shows the bound of IPC under different level of noise, but that doesn't explain the trend in Figure 6,7, i.e., how the accuracy change with noise rate with different IPC. Is there any insight or try on predicting this trend, like a scaling law for data distillation?

---

### Meta-Review · Area_Chair_52yP · 2025-12-11

**Summary:**

The paper surfaced an intriguing claim: distilled datasets can denoise high-noise labels. However, the empirical evidence is narrow, lacks model diversity, omits direct comparisons to standard noisy-label methods, and gives limited analysis of asymmetric noise where performance degrades. Theoretical bounds are interesting but insufficiently stress-tested --- reviewers sought quantitative validation and practical guidance, which remain open. Thus, the contributions are not yet enough to go beyond the ICLR acceptance bar.

**Reviewer Concerns:**

**9Gpj** whether theory explains accuracy-noise trends across IPC and whether a predictive “scaling law” exists.

**cF8n** quantitative tests of Corollaries I–III, clearer link between semantic diversity and asymmetric-noise bounds, details of flip probabilities, missing settings in Fig. 5, broader datasets/methods.

**2b5n** limited applicability, small scale, missing comparisons to noise-robust baselines.

**upv1** no modern architectures, larger/imbalanced data, failure under asymmetric noise, augmentation baselines, sample semantics, cross-architecture transfer, compute cost.

No rebuttal submitted by the authors, and all concerns are still outstanding.

**Reviewer Scores:**

No reason at all to increase their scores.

---

### Decision · Program_Chairs · 2026-01-26

Reject